# Fault Network Reconstruction using Agglomerative Clustering: Applications to South Californian Seismicity

Yavor Kamer[1,a], Guy Ouillon[2], Didier Sornette[1]

[1]ETH Zurich, Switzerland
[2]Lithophyse, Nice, France
[a]now at: RichterX.com

*Correspondence to*: Yavor Kamer (yaver.kamer@gmail.com)

## Abstract

In this paper we introduce a method for fault network reconstruction based on the 3D spatial distribution of seismicity. One of the major drawbacks of statistical earthquake models is their inability to account for the highly anisotropic distribution of seismicity. Fault reconstruction has been proposed as a pattern recognition method aiming to extract this structural information from seismicity catalogs. Current methods start from simple large scale models and gradually increase the complexity trying to explain the small scale features. In contrast the method introduced here uses a bottom-up approach that relies on initial sampling of the small scale features and reduction of this complexity by optimal local merging of substructures.

First, we describe the implementation of the method through illustrative synthetic examples. We then apply the method to the probabilistic absolute hypocenter catalog KaKiOS-16, which contains three decades of South Californian seismicity. To reduce data size and increase computation efficiency, the new approach builds upon the previously introduced catalog condensation method that exploits the heterogeneity of the hypocenter uncertainties. We validate the obtained fault network through a pseudo prospective spatial forecast test and discuss possible improvements for future studies. The performance of the presented methodology attests the importance of the non-linear techniques used to quantify location uncertainty information, which is a crucial input for the large scale application of the method. We envision that the results of this study can be used to construct improved models for the spatio-temporal evolution of seismicity.

## 1. Introduction

Owing to the continuing advances in instrumentation and improvement of seismic networks coverage, earthquake detection magnitude thresholds have been decreasing while the number of recorded events is increasing. As governed by the Gutenberg-Richter law, the number of earthquakes above a given magnitude increases exponentially as the magnitude is decreased (Ishimoto and Iida, 1939; Gutenberg and Richter, 1954). Recent studies suggest that the Gutenberg-Richter law might hold down to very small magnitudes corresponding to interatomic-scale dislocations (Boettcher et al., 2009; Kwiatek et al., 2010). This implies that there is practically no upper limit on the amount of seismicity we can expect to record as our

instrumentation capabilities continue to improve. Although considerable funding and research efforts are being channeled into recording seismicity, when we look at the uses of the end product (i.e. seismic catalogs) we often see that the vast majority of the data (i.e. events with small magnitudes) are not used in the analyses. For instance, probabilistic seismic hazard studies rely on catalogs containing events detected over long terms, which increases the minimum magnitude that can be considered due to the higher completeness magnitude levels in the past. Similarly, earthquake forecasting models are commonly based on the complete part of the catalogs. For instance, in their forecasting model, (Helmstetter et al., 2007) use only $M>2$ events, which corresponds to only ~30% of the recorded seismicity. The forecasting skills of the current state-of-the-art models can well be hindered not only due to our limited physical understanding of earthquakes, but also due to this data censoring.

In this conjecture, fault network reconstruction can be regarded as an effort to tap into this seemingly neglected but vast data source, and extract information in the form of parametric spatial seismicity patterns. We are motivated by the ubiquitous observations that large earthquakes are followed by aftershocks that sample the main rupturing faults, and conversely that these faults become the focal structures of following large earthquakes. In other words, there is a relentless cycle as earthquakes occur on faults that themselves grow by accumulating earthquakes. By using each earthquake, no matter how big or small, as a spark in the dark, we aim to illuminate and reconstruct the underlying fault network. If the emerging structure is coherent, it should allow us to better forecast the spatial distribution of future seismicity and also to investigate possible interactions between its constituent segments.

The paper is structured as follows. First, we give an overview of recent developments in the field of fault network reconstruction and spatial modeling of seismicity. In Section 2, we describe our new clustering method and demonstrate its performance using a synthetic example. In Section 3, we apply the method to the recently relocated southern California catalog KaKiOS-16 (Kamer et al., 2016) and discuss the obtained fault network. In Section 4, we perform a pseudo-prospective forecasting test using four years of seismicity that was recorded during 2011-2015 and was not included in the KaKiOS-16 catalog. In the final Section, we conclude with an outlook on future developments.

## 2. Recent developments in fault reconstruction

In the context the work presented here, we use the term "fault" as a three-dimensional geometric shape or kernel optimized to fit observed earthquake hypocenters. Fault network reconstruction based on seismicity catalogs was introduced by (Ouillon et al., 2008). The authors presented a dynamical clustering method based on fitting the hypocenters distribution with a plane, which is then iteratively split into an increasing number of subplanes to provide better fits by accounting of smaller scale structural details. The method uses the overall location uncertainty as a lower bound of the fit residuals to avoid over fitting. (Wang et al., 2013) made further improvements by accounting for the individual location uncertainties of the events and introducing motivated quality evaluation criteria (based, for instance, on the agreement of the planes orientations with the events focal mechanisms). (Ouillon and Sornette, 2011) proposed an alternative method based on probabilistic mixture modeling (Bishop, 2007) using 3D Gaussian kernels. This method introduced notable improvements, such as the use

of an independent validation set to constrain the optimal number of kernels to explain the data (i.e. model complexity) and
diagnostics based on nearest-neighbors tetrahedra volumes to eliminate singular clusters that cause the mixture likelihood to
diverge. While our method is inspired by these studies, and in several aspects builds upon their findings, we also note an
inherent drawback of the iterative splitting approach that is common to all the previously mentioned methods. This can be
observed when an additional plane (or kernel), introduced by splitting, fails to converge to the local clusters and is instead
attracted to the regions of high horizontal variance (see Figure 1 for an illustration in the case of Landers' seismicity).

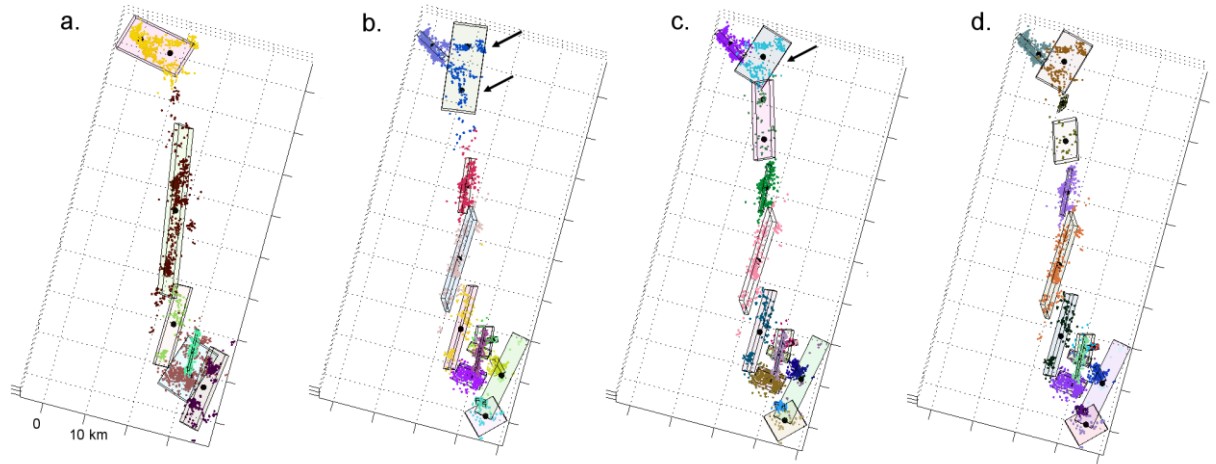


**Figure 1** Iterative splits on the 1992 Landers aftershock data. Points with different colors represent seismicity associated with each plane.
Black dots show the center points of the planes resulting from the next split. Notice how in steps b. to c. step the planes fail to converge to
the local branches (shown with arrows), and the method prefers to introduce a horizontal plane to fit a more complex local pattern.
This deficiency has motivated us to pursue a different concept. Instead of starting with the simplest model (i.e. a
single plane or kernel) and increasing the complexity progressively by iterative splits, we propose just the opposite: start at
the highest possible complexity level (as many kernels as possible) and gradually converge to a simpler structure by iterative
merging of the individual substructures. In this respect, the new approach can be regarded as a "bottom-up" while the
previous ones are "top-down" approaches.
**3. The agglomerative clustering method**
**3.1. Method description**
The method shares the basic principles of agglomerative clustering (Rokach and Maimon, 2005) with additional
improvements to suit the specifics of seismic data, such as the strong anisotropy of the underlying fault segments. We
illustrate the method by applying it to a synthetic dataset obtained by sampling hypocenters on a set of five plane segments,
and potentially adding uncorrelated background points which are uniformly distributed in the volume (see Figure 2). The
implementation follows the successive steps described below:
i) For a given dataset featuring $N$ hypocenters, we first construct an agglomerative hierarchical cluster (AHC) tree
based on Ward's minimum variance linkage method (Ward, 1963). Such a tree starts out with a cluster for each data-point
(i.e., with zero variance) and then progressively branches into an incrementally decreasing number of clusters (see Figure 2
c,d). At any step, the merging of two clusters is based on a criterion involving the minimum distance $D_w$ criterion given by:

$$D_w\left(C_i, C_j\right) = \sum_{x \in C_{ij}} \left(x - r_{ij}\right)^2 - \sum_{x \in C_i} \left(x - r_i\right)^2 - \sum_{x \in C_j} \left(x - r_j\right)^2 \tag{1}$$

In this equation, $C_{ij}$ is the cluster formed by merging clusters $C_i$ and $C_j$, $x$ represents the set of hypocenters, and $r$ (with
proper subscript) is the centroid of each cluster. Hence, clusters $i$ and $j$ are merged if the sum of squares in Eq. (1) is
minimized after they are merged into a single cluster $ij$. The number of branches in the tree is thus reduced by one, and the
remaining clusters are used to decide which ones will be merged at the next iteration. This merging of clusters/branches
continues until there remains only a single cluster. "Cutting" the AHC tree at the $D_w$ level corresponding to the desired
number of branches allows one to choose the number of clusters (from 1 to $N$) used to represent the original dataset. While
there are many different linkage methods and distance metrics, here we have chosen to use Ward's criterion because it
produces clusters with regular sizes. This is important for the atomization procedure as we want clusters to have similar
potentials to merge and grow into bigger structures.
ii) Since our goal is to obtain a fault network where segments are modeled by Gaussian kernels, we begin by
estimating how many such kernels can be constructed with the clusters featured in the AHC tree. At its most detailed level
($N$ clusters) no such kernel exists as they would collapse on each data point, becoming singular. At the next level ($N-1$
clusters), we have the same problem. We thus incrementally reduce the level, traversing AHC tree, until we get a first cluster
featuring 4 hypocenters, which defines the first non-singular cluster. We then continue our traverse along the tree down
replacing each cluster having more than 4 points by a Gaussian kernel. At each level on the tree, we count the number of
these non-singular Gaussian kernels. The results are illustrated on Figure 2b where we consider two cases: first considering
only the 5 planes, the second one including a set of uniformly distributed background points. In the first case, we see that the
maximum number of Gaussian kernels (76) is obtained when we cut the tree so that the total number of clusters is 117. In the
second case, in the presence of background points, the maximum number of Gaussian kernels (77) is obtained when we cut
the tree at a level of 214 clusters. We refer to this maximum number is as the "holding capacity" of the dataset, and the
corresponding configuration defines the starting point of the following iterative and likelihood-based clustering algorithm.
The process of finding this optimum set of initial Gaussian proto-clusters (all containing more than 4 points) is coined as
"atomization".

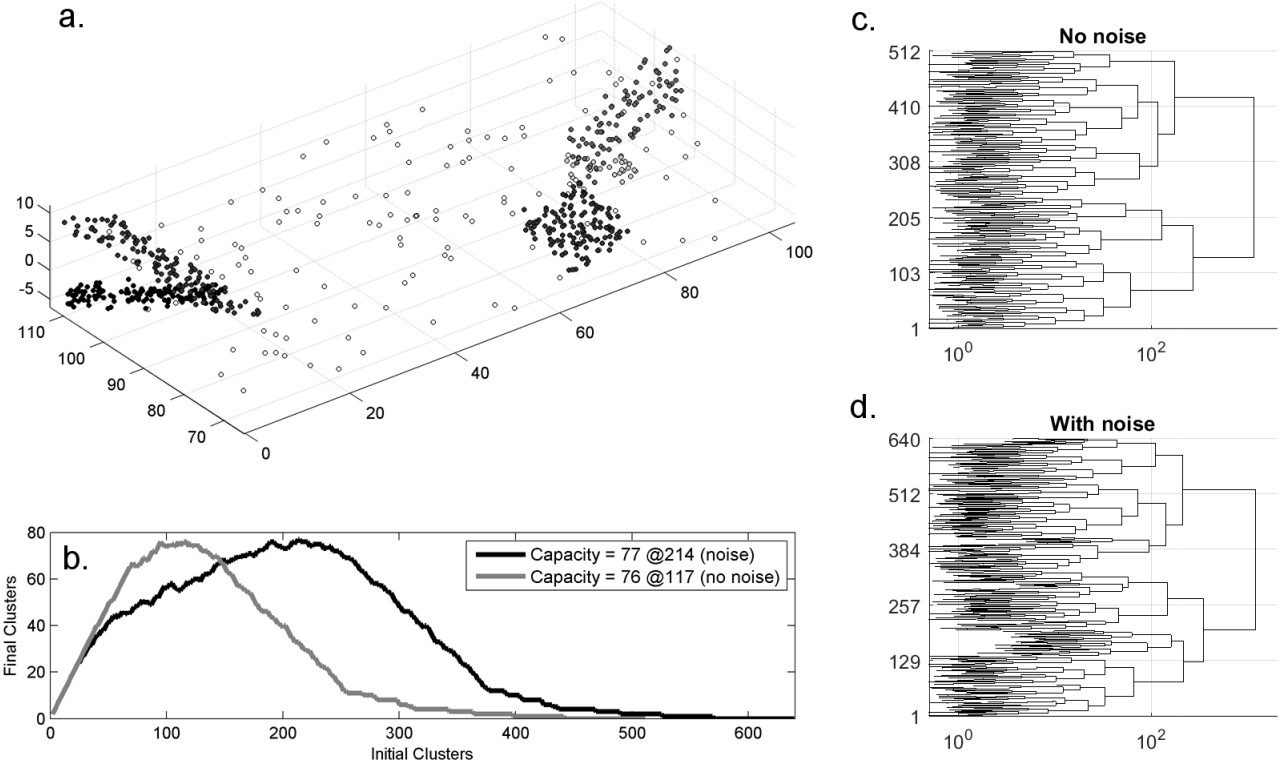


**Figure 2**  a) Synthetic fault network with 640 points created by uniform sampling of 5 faults, each shown with a different shade according
to its total number of points. Empty circles represent the %20 uniformly random background points. b) Determination of the holding
capacity (see main text) for the case with and without background points. c-d) Dendrograms showing the agglomerative hierarchical
cluster tree for the data with no noise (c) and with noise (d). The horizontal length of each branch is the minimum distance $D_w$ (see Eq.1)
joining two clusters
iii) Once we determine the holding capacity, all points that are not associated with any Gaussian kernel are assigned
to a uniform background kernel that encloses the whole dataset. The boundaries of this kernel are defined as the minimum
bounding box of its points. The uniform spatial density of this background kernel is defined as number of points divided by
the volume (see Figure 3). The Gaussian kernels together with the uniform background kernel represent a mixture model
where each kernel has a contributing weight proportional to the number of points that are associated with it (Bishop, 2007).
This representation facilitates the calculation of an overall likelihood and allows us to compare models with different
complexities using the Bayesian Information Criteria (BIC) (Schwarz, 1978) given by:

$$BIC = -\sum_{i}^{N}\log(L) + \tfrac{k}{2}\log(N)$$
(2)

where $L$ is the likelihood of each data point, $k$ is the number of free parameters of the mixture model and $N$ is the total
number of data points. The value of $k$ is calculated as $k=10N_C-1$ (where $N_C$ is the number of kernels in the mixture) since
each kernel requires 3 (mean vector) + 6 (covariance matrix) + 1 (weight) = 10 free parameters. The same parameterization
is also used to describe the background kernel, which is a uniformly dense cuboid with a size and orientation prescribed by
its covariance matrix. The number of free parameters ($k$) is reduced by 1 because the weights have to sum to unity and hence
knowing $N_C-1$ of them is sufficient.

iv) At the holding capacity, the representation with the large number of kernels is likely to constitute an overfitting

model for the data set. Therefore, we iteratively merge pairs of the Gaussian kernels until an optimal balance between fitness
and model complexity is reached. We use the measure of information gain in terms of BIC to select which pair of kernels to
merge. For any given pair of Gaussian kernels, the BIC gain resulting from their merger is calculated using Equation (3)
where $L_{int}$ is the likelihood of each data-point for the initial (unmerged) model and $L_{mrg}$ is the likelihood in the case where the
two candidate clusters are merged:

$$BIC_{Gain} = BIC_{int} - BIC_{mrg}$$

$$BIC_{int} = -\sum_{i}^{N} \log(L_{int}) + \tfrac{k}{2}\log(N)$$

$$BIC_{mrg} = -\sum_{i}^{N} \log(L_{mrg}) + \tfrac{k-10}{2}\log(N) \tag{3}$$

$$BIC_{Gain} = \sum_{i}^{N} \log(L_{mrg}) - \sum_{i}^{N} \log(L_{int}) + 5\log(N)$$

Notice that each merging of a pair of kernels decreases $k$ by 10, thus a given merger can be considered only if the reduction
of the penalty term is greater than the decrease of likelihood (i.e. $BIC_{Gain}>0$).

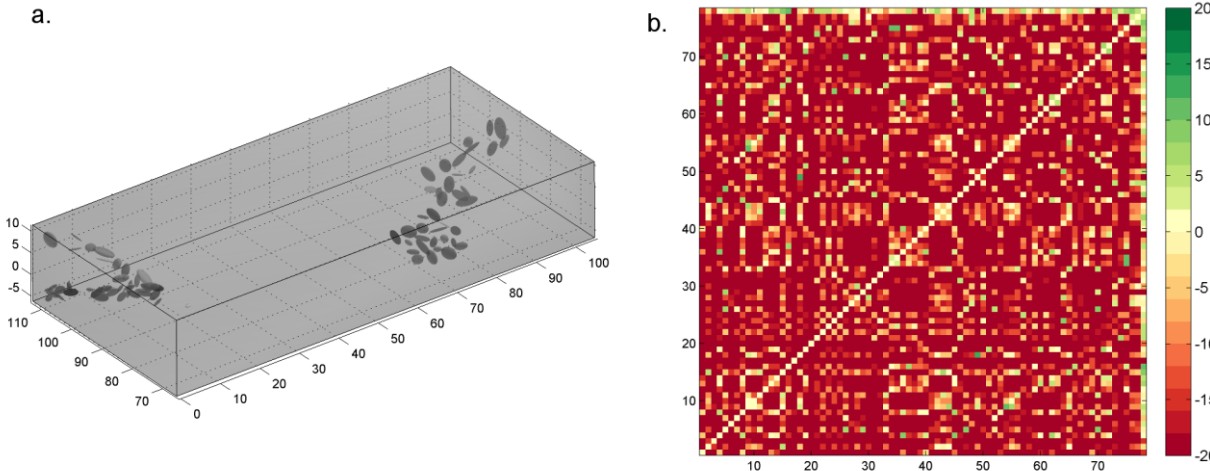


**Figure 3** a) The initial *protoclusters* for the synthetic dataset given in Figure 2a. Notice that the number of clusters (78) includes the uniform background kernel as well. b) The *BIC* gain matrix calculated for all possible merging of pairs of kernels.

Using this formulation, we calculate a matrix where the value at the intersection of $i^{th}$ row and $j^{th}$ column corresponds to the BIC gain for merging clusters $i$ and $j$. We merge the pair with the maximum BIC gain and then re-estimate the matrix since we need to know the BIC gains of the newly formed cluster. At each step, the complexity of the model is reduced by one cluster, and the procedure continues until there is no merging yielding a positive BIC gain. Figure 3b shows such a BIC gain matrix calculated for the initial model with 77 clusters. Notice that a Gaussian cluster is not allowed to merge with the background kernel. The $BIC_{Gain}>0$ criteria, which essentially drives and terminates the merging process, is similar to a likelihood ratio test (Neyman and Pearson, 1933; Wilks, 1938) with the advantage that the models tested do not need be nested.

The computational demand of the BIC gain matrix increases quadratically with the number of data points. To make our approach feasible for large seismic datasets, we introduce a preliminary check that considers clusters as candidates for merging only if they are overlapping within a confidence interval of $\sigma\sqrt{12}$ in any of their principal component directions. The factor $\sqrt{12}$ is derived from the variance of an hypothetical uniform distribution over a planar surface (for details see (Ouillon et al., 2008)).

During all steps of the merging procedure, the data points are in the state of *soft clustering*, meaning that they have a finite probability to belong to any given kernel. A deterministic assignment can be achieved by assigning each point to the kernel that provides the highest responsibility (as per the definition of a mixture model), which is referred to as *hard clustering*. This dichotomy between stochastic and deterministic inference gives rise to two different implementations for the merging criteria: 1) *local* criterion: considering only the two candidate clusters and the data-points assigned to them through hard-clustering and 2) *global* criterion: considering the likelihood of all data-points for all clusters. In essence, the *local* criterion

tests the information gain for the case of two kernels versus one kernel on a subset, whereas the global criterion considers $N_c$
versus $N_c$-1 kernels on the whole mixture and dataset. Figure 4 shows the resulting final reconstructions for the two criteria.

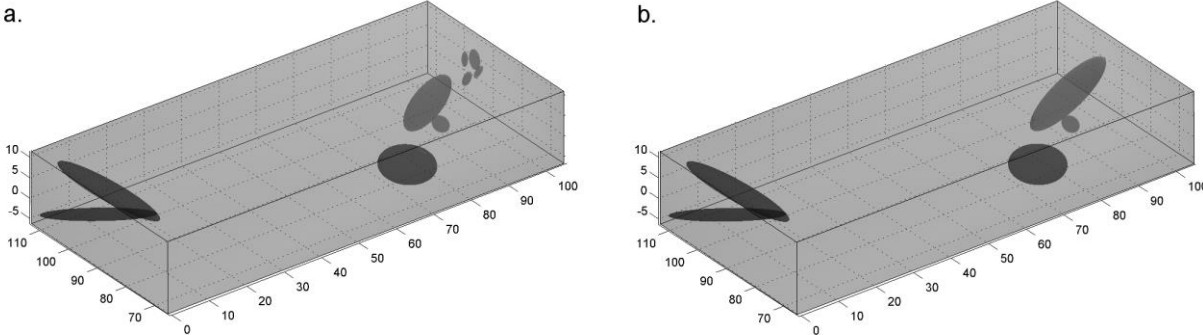

**Figure 4** The final models obtained using the local (a) and global (b) merging criteria for the dataset presented on Figure 2. The number of
clusters, including the uniform background kernel, is 11 and 6 for the local and global criteria respectively.
For this synthetic dataset, we observe that both the local and global criteria converge to a similar final structure. The global
criterion yields a model with the same number of clusters as the input synthetic, while the local criterion introduces four
additional clusters in the under-sampled part of one of the faults. For most pattern recognition applications that deal with a
robust definition of noise and signal, the global criterion may be the preferred choice since it is able to recover the true
complexity level. However, since this method is indended for natural seismicity, we also see a potential in the local criterion.
For instance, consider the case where two fault segments close to each other are weakly active and thus have a low spatial
density of hypocenters compared to other distant faults that are much more active. In that case, the global criterion may
choose to merge the low-activity faults, while the local criterion may preserve them as separate.
**3.2. Sensitivity analysis**
In order to gain insight about the sensitivity and the robustness of the proposed method, we conduct a more elaborate
synthetic test. We generate a set of 20 randomly oriented planes with their attributes varying in the following ranges: strike
angle -90° to 90°, dip angle 45° to 90°, length 20 to 40 km, width 5 to 15 km. The fault planes span a region with the
dimensions of 220 x 150 x 30 km. Each fault plane is sampled randomly with an increasing number of points; starting from
0.1 point/km2 going up to 2 points/km$^2$ in 15 steps, producing sets with a total number of points in the range of 609 to
14,475. We also consider three different uniform background noise levels at 5%, 10% and 20% yielding a total of 45
synthetic sets. We apply our clustering method to each of these sets and report the resulting performance using the Rand
index. The Rand index measures the similarity between different clusterings and is expressed by the following equation

$$R = \frac{2(a+b)}{n(n-1)} \qquad (4)$$


where $a$ is the number of pairs that are in the same cluster in the two clusterings, $b$ is the number of pairs that are in different
clusters in the two clusterings and $n$ is the number of points in the dataset. A Rand index of 1 indicates total match between
the two groupings while a value of 0 indicates that all pairs are in disagreement. In our case, we are comparing the ground
truth clustering, which is given by the 20 fault planes and the uniform background, and the clustering obtained by our
method. Figure 5a shows the Rand index obtained using the *local* and the *global* criteria as a function of increasing sampling
density for the three levels of background noise. As mentioned earlier, the performance of the *global* criterion is better than
the *local* one, which degrades with increasing density as the method start introducing additional clusters. The Rand index of
the *global* criterion saturates around 0.95 and starts decreasing as the density increases above ~1.25 points/km². This
saturation can be explained by the fact that additional Gaussian kernels are needed to fit the sharp corners of the rectangular
planes as they become more pronounced with increased sampling. We can make an analogy with the Fourier series
expansion of a square wave, where more terms are needed to fit the sharp edges. In our case, these additional terms (i.e.
Gaussian kernels) increase the complexity and cause the Rand index to drop. To confirm this we repeat the synthetics by
sampling Gaussian kernels with the eigenvectors corresponding to the rake and dip, and eigenvalues corresponding to the
length, width and thickness of the rectangular planes. The results are shown in Figure 5b where we see no drop off in the
Rand index.

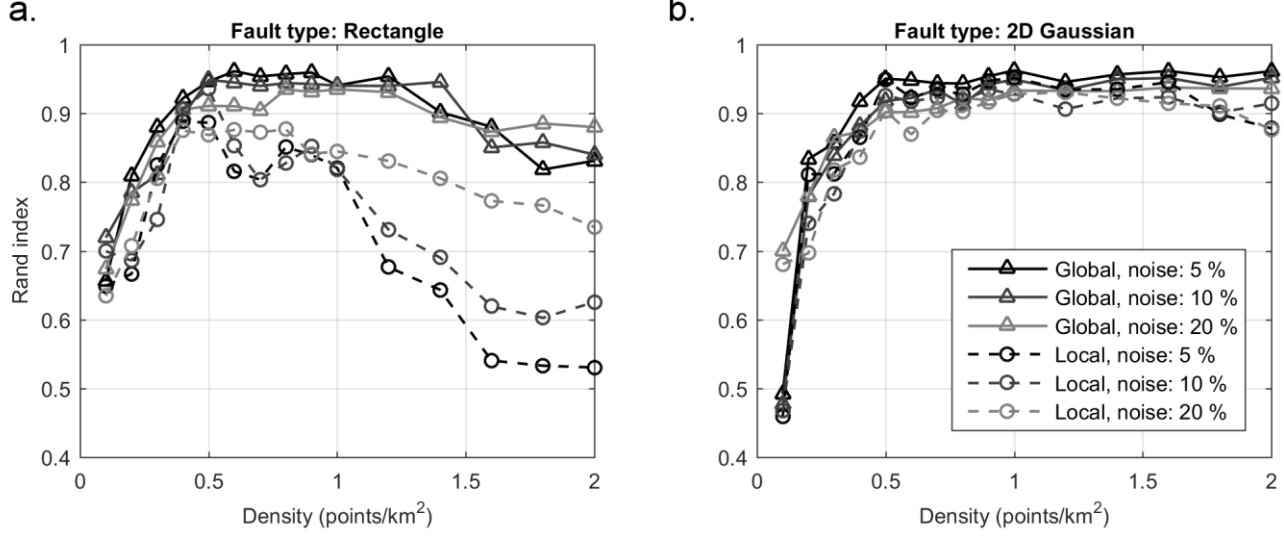


**Figure 5** Clustering similarities between ground truth synthetic dataset and method results quantified by the Rand index. *Global* and *local*
merging criteria are shown as solid and dashed lines respectively. Background noise amplitude is shown as shades of gray. Results for
ground truth sampled from a) rectangular fault planes b) elliptic Gaussian kernels with similar dimensions.
These synthetics indicate that the method is robust in the presence of uniform background noise and that it is able to
recover structures that are sufficiently sampled. In the presented case, the performance saturates around 0.5
points/km$^2$, however this value can change based on the particular setting. For instance, if faults are very closely
spaced and intersecting, higher sampling may be needed. On the other hand, if the structures are isolated, similar
performance can be achieved at lower sampling. The MATLAB code used for generating the synthetics and evaluating
the reconstruction's Rand index is provided. Users may prefer to create synthetic cases that are informed by the
properties of the actual data they are working on (such as numbers of points, spatial extend, etc.)

## 4. Application to seismicity

In this section, we apply our method to observed seismicity data. For this purpose, we use the KaKiOS-16 catalog
(Kamer et al., 2016) that was obtained by probabilistic absolute location of nearly 479,000 Southern Californian events
spanning the time period 1981-2011. We consider all events, regardless of magnitude, as each event samples some part of
the fault network. Before tackling this vast dataset, however, we first consider the 1992 Landers sequence as a smaller
dataset to assess the overall performance and computational demands.

### 4.1. Small Scale application to the Landers aftershocks sequence

We use the same dataset as (Wang et al., 2013) that consists of 3,360 aftershocks of the 1992 Landers earthquake.
The initial atomization step produces a total of 394 proto-clusters that are iteratively merged using the two different criteria
(local and global). The resulting fault networks are given in Figure 6 together with the fault traces available in the
Community Fault Model of southern California (Plesch et al., 2007). Comparing the two fault networks, we observe that the
local criterion provides a much detailed structure that is consistent with the large scale features in the global one. We also
observe that, in the southern end, the global criterion produces thick clusters by lumping together small features with
seemingly different orientations. These small scale features have relatively few points and thus low contribution to the
overall likelihood. The global criterion favors these mergers to reduce the complexity penalty in Equation (2), which scales
with the total number of points. In the local case, however, because each merger is evaluated considering only the points
assigned to the merging clusters, the likelihood gain of these small scale features can overcome the penalty reduction and
they remain unmerged. It is also possible to employ metrics based on consistency of focal mechanism solutions to evaluate
the reconstructed faults. For a detailed application of such metrics the reader is referred to the detailed work by Wang et
al.(2013). In this study, since we do not have focal mechanism solutions for our target catalog, we focus on information
criteria metrics and out of sample forecast tests.

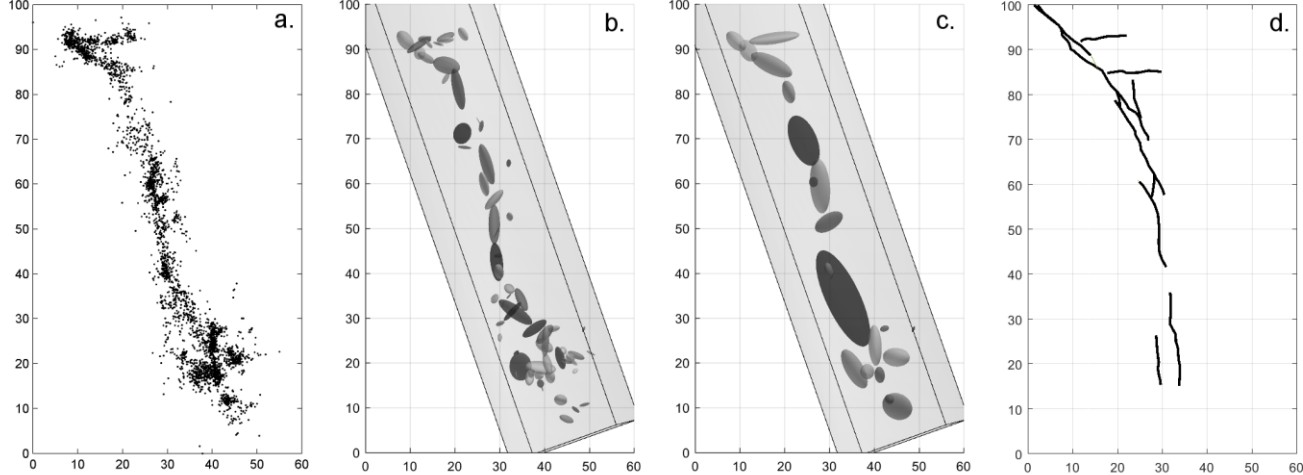

**Figure 6** a) Top view of the 1992 Landers aftershocks. Fault networks obtained from these events using the local (b) and global (c) merging criterion, each resulting in 70 and 22 clusters respectively. d) Fault traces obtained from the Community Fault Model of southern California

Our second observation is that the background kernel attains a higher weight of 11% using the local criterion compared to the global one yielding only 5%. Keeping in mind that both criteria are applied on the same initial set of proto-clusters, and that there are no mergers with the background kernel, we argue that the difference between the background weights is due to density differences in the tails of the kernels. We investigate this in Figure 7 for the simple 1D case considering mergers between two boxcar functions (analogous for planes in 3D) approximated with Gaussian functions. We observe that the merged Gaussian has higher densities in its tails compared to its constituents. The effect is amplified when the distance between the merging clusters is increased (Figure 7b). Hence, in the local case, the peripheral points are more likely to be associated with the background kernel due to the lower densities at the tails of the small, unmerged clusters.

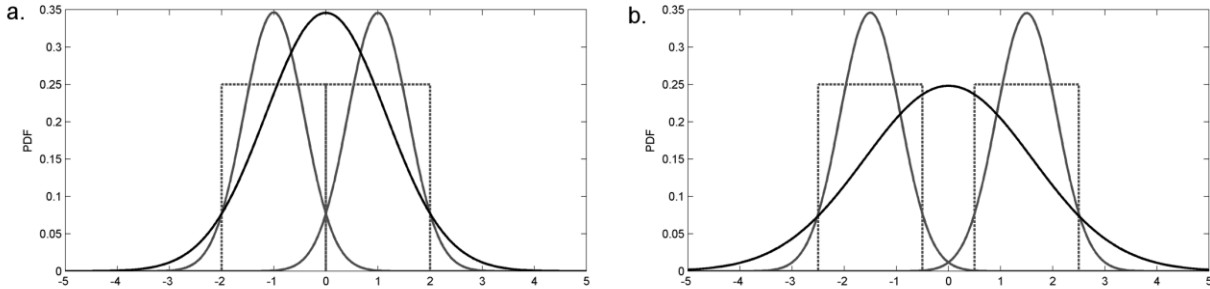

**Figure 7** Two uniform distributions (dotted gray lines), their Gaussian approximations (solid gray lines) and the Gaussian resulting from their merger (solid black line). Notice that the joint Gaussian has higher densities at the tails compared to its constituents.

Another important insight from this sample case was regarding the feasibility of a large scale application. As
pointed out here and in previous studies (Ouillon and Sornette, 2011; Wang et al., 2013), the computational demand for such
pattern recognition methods increases rapidly with the number of data-points. The Landers case with 3,360 points took ~5
minutes on a 4-core, 2.2GHz machine with 16GB memory. Considering that our target catalog is nearly ~145 times larger, a
quadratic increase would mean an expected computation time of more than two months. Even with a high performance
computing cluster, we would have to tackle memory management and associated overhead issues. Although technically
feasible, pursuing this path would limit the use of our method only to the few privileged with access to such computing
facilities. In a previous work we proposed a new solution called "catalog condensation", that uses the location uncertainty
estimates to reduce the length of a catalog while preserving its spatial information content (Kamer et al., 2015). In the
following section, we will detail how we applied this method to the KaKiOS-16 catalog in order to make the clustering
computations feasible.
**4.2. Condensation of the KaKiOS-16 catalog**
The condensation method reduces the effective catalog length by first ranking the events according to their location
uncertainty and then successively condensing poorly located events onto better located ones (for detailed explanation see
Kamer et al., 2015). The initial formulation of the method was developed considering the state of the art catalogs of the time.
Location uncertainties in these catalogs are assumed to be normally distributed and hence expressed either in terms of a
horizontal and vertical standard deviation, or with a diagonal 3x3 covariance matrix. With the development of the KaKiOS-
16 catalog, we extended this simplistic representation to allow arbitrarily complex location PDFs to be modeled with
mixtures of Gaussians. Such mixture models, consisting of multiple Gaussian kernels, were found to be the optimal
representation for 81% percent of the events, which required an average of 3.24 Gaussian components (the rest was
optimally modeled using a single Gaussian kernel). Therefore we first needed to generalize the condensation methodology,
which was initially developed for single kernels, to accommodate the multiple kernel representation. In the original version,
all events are initiated with equal unit weights. They are then ranked according to their isotropic variances and weights are
progressively transferred from the high variance to the low variance events according to their overlap. In the generalized
version, each event is represented by a number of Gaussian kernels that are initiated with their respective mixture weight (0-
1). All kernels are then ranked according to their isotropic variance and the weights are transferred as in the original method
with the additional constraint that weight transfers between kernels of the same event are not allowed (see Figure 8a, b). This
constraint is motivated by the fact that the kernels representing each event's location PDF are already optimized. Thus a
weight transfer between those can lead only to a sub-optimal location representation.

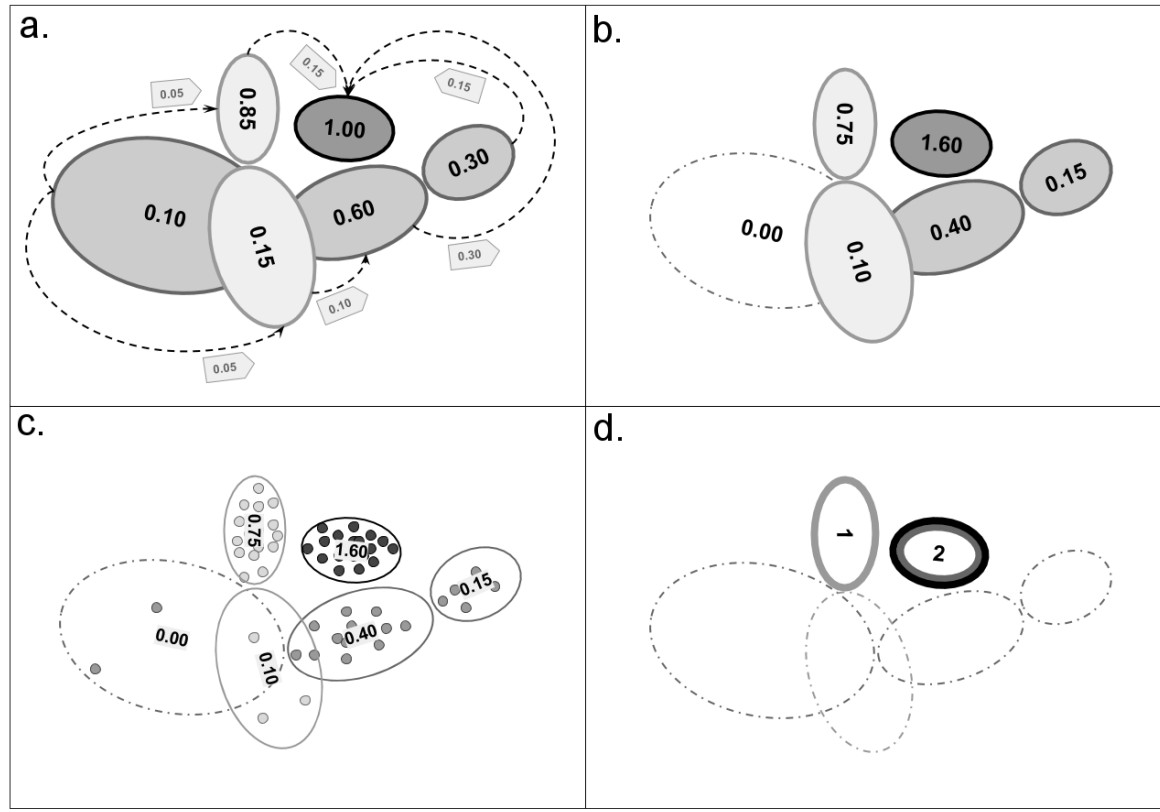


**Figure 8** Idealized schematic representations of 3 events with 1,2 and 3 Gaussian kernels each a) Condensation: each event is represented
by a different shade, weight transfer is represented by the arrows; notice that there are no intra-event weight transfers b) Final condensed
catalog: the total weight sum is preserved, one component is discarded. c) Sampling of the event PDFs: this step is done on the original
catalog d) Each event is assigned to the condensed kernel that provides the maximum likelihood for most of its sampled points; three
events are assigned to two condensed kernels.
The KaKiOS-16 catalog contains 479,056 events whose location PDFs are represented by a total of 1,346,010
Gaussian components (i.e. kernels). Condensation reduces this number to 600,463 as weights from events with high variance
are transferred to better located ones. Nevertheless, in Figure 9 we see that nearly half of these components amount to only
10% of the total event weight. The computation time scales with the number of components, while the information content is
proportional to number of events. Hence the large number of components amounting to a relatively low number of events
would make the computation inefficient. A quick solution could be to take the components with the largest weights
constituting 90% or 95% of the total mass, mimicking a confidence interval. Such a "solution" would depend on the arbitrary
cutoff choice and would have the potential to discard data that may be of value for our application.

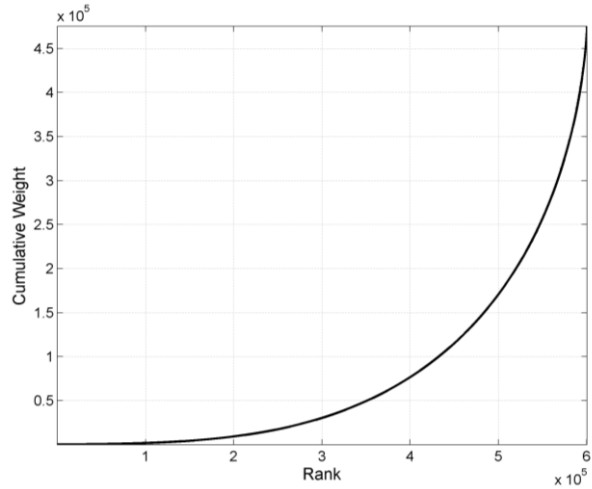


**Figure 9** Cumulative weights of the 600,463 condensed KaKiOS-16 components representing a total of 479,056 events. The components are ranked according to increasing weights.

We can avoid such an arbitrary cut-off by employing the fact that the condensed catalog is essentially a Gaussian
mixture model (GMM) representing the spatial PDF of earthquake occurrence in South California. We can then, in the same
vein as the hard clustering described previously, assign each event to its most likely GMM component (i.e. kernel). If we
consider each event individually, the most likely kernel would be the one with the highest responsibility. However, for a
globally optimal representation we need to find the best representative kernel for each event among all other kernels. To do
this, we sample the original (uncondensed) PDF of each event with 1000 points and then calculate the likelihood of each
sample point with respect to all the condensed kernels. The event is assigned to the kernel that provides the maximum
likelihood for the highest number of sample points (see Figure 8c,d). As a result of this procedure, the 479,056 events are
assigned to 93,149 distinct kernels. The spatial distribution of all the initial condensed kernels is given in Figure 10a, while
the kernels assigned with at least one event after the hard clustering are shown in Figure 10b. Essentially, this procedure can
be viewed as using the condensed catalog as a prior for the individual event locations. The use of accumulated seismicity as
a prior for focusing and relocation has been proposed by Jones and Stewart (1997) and investigated in detail by Li et al.
(2016). We can see the effect of this strategy more clearly in Figure 8, where starting from 3 different events in the catalog
(Figure 8a), we finally converge to only 2 different final locations (Figure 8d).

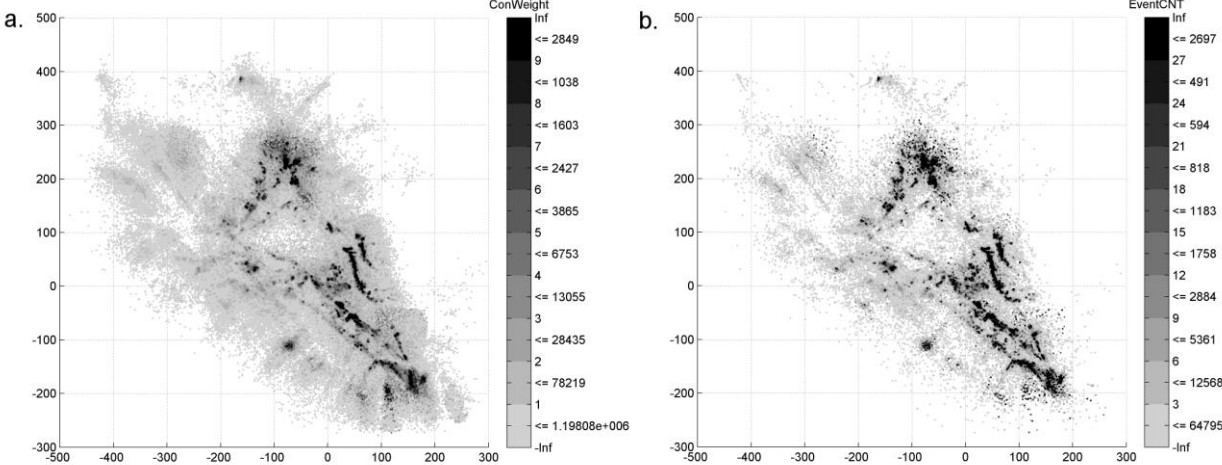

**Figure 10** a: Mean locations of condensed 600,463 Gaussian components shaded according to their weights. b: The same components shaded according to the total number of events assigned to them after the maximum likelihood assignment

### 4.3. Large scale application to Southern California

In previous works, we concluded that the spatial distribution of southern California seismicity is multifractal, i.e. it is an inhomogeneous collection of singularities (Kamer et al., 2015, 2016). The spatial features in Figure 10 can be seen as expressions of these singularities. Since we are interested in the general form of the fault network rather than the second order features (e.g. inhomogeneous seismicity rates along the same fault) we consider all the centers of all 93,149 kernels as individual points, effectively disregarding their weights. Considering the weight of each kernel would result in more complex structure with singularities that can be associated with the fractal slip distribution of large events (Mai and Beroza, 2002) modulated through the non-uniform network detection capabilities. Thus, by disregarding the kernel weights we are considering only the potential loci of earthquakes, not their activity rates.

Another important aspect, in the case of such a large scale application, is the uniform background kernel. The assumption of a single background kernel defined as the minimum bounding box of the entire dataset seems to be suitable for the case of Landers aftershocks, however it becomes evident that for whole Southern California such a minimum bounding box would overestimate the data extent (covering aseismic offshore areas) and would thus lead to an underestimated density. In addition, one can also expect the background density to vary regionally in such large domains. We thus extend our approach by allowing for multiple uniform background kernels. For this purpose, we make use of the AHC tree that is already calculated for the atomization of the whole dataset. We then cut the tree at a level corresponding to only a few clusters (5 or 30 in the following application), which allows to divide the original catalog into smaller subcatalogs represented by each cluster. Each of these subsets is then atomized individually yielding its own background kernel. The atomized subsets are then brought together, to be progressively merged. Naturally, we have no objective way of knowing how many background kernels a dataset may feature. However, in various synthetic tests, involving cuboid backgrounds

with known densities, we observe that inflating this number has no effect on the recovered densities, whereas a too low value
causes underestimation. Apart from this justification, we are motivated to divide this large dataset into subsets for purely
computational reasons as this allows for improved parallelization and computational efficiency.

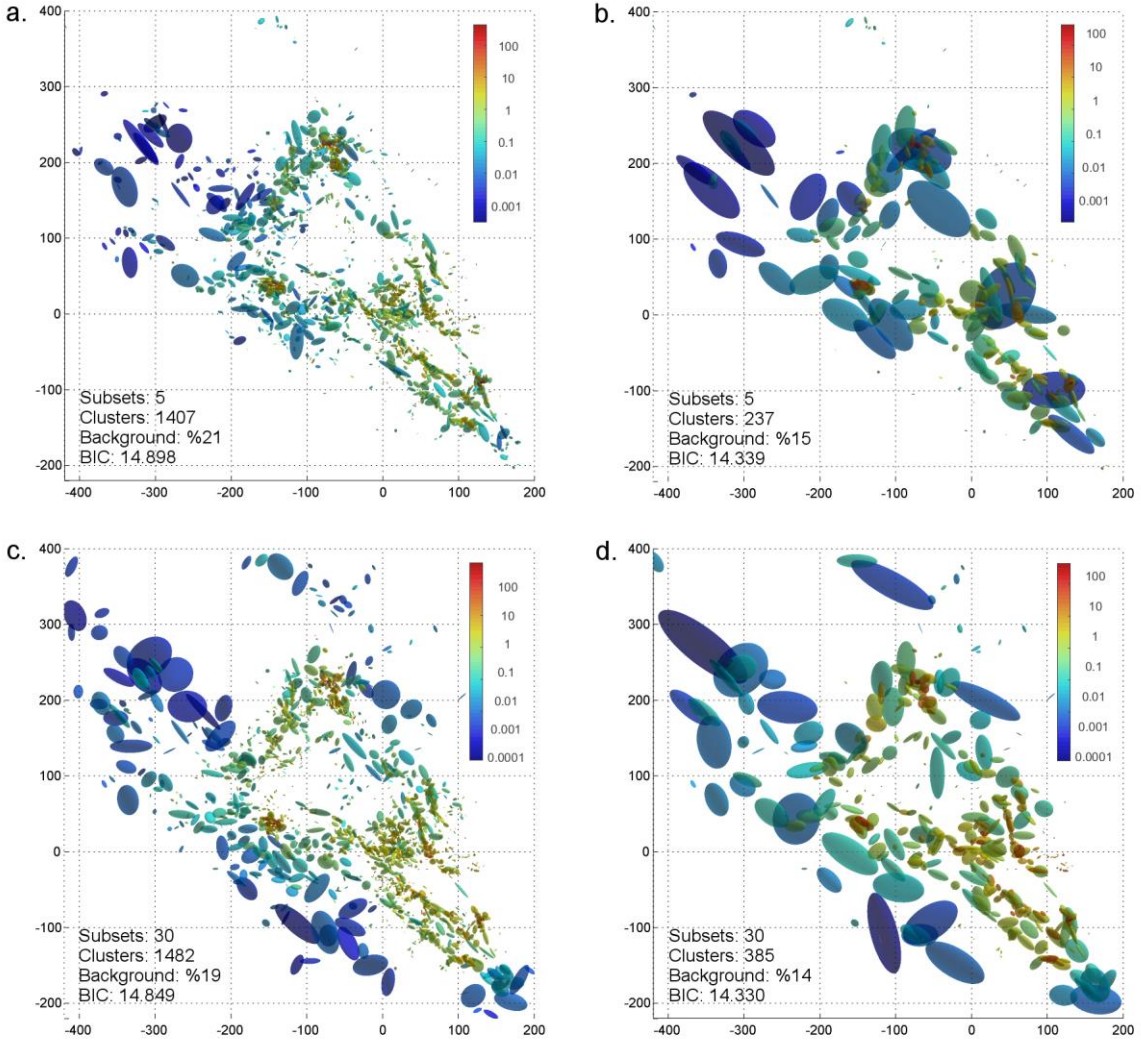


**Figure 11** Fault network reconstructions for the KaKiOS-16 catalog. Top row shows results for the case of 5 initial subsets with (a) local
and (b) global merging criterion. Bottom row shows the (c) local and (d) global merging criterion for 30 initial subsets. The number of
clusters, background weight and BIC per data point is given in the insets. Clusters are colored according their density (data point per km$^3$)
where the volume is estimated as the product of standard deviations along the principal component axes.

Figure 11 shows the two fault networks obtained for two different initial settings: using 5 and 30 subsets. For each
choice, we show the results of the local and global criterion; the background cuboids are not plotted to avoid clutter. Our
immediate observation is related to the events associated with the 1986 Oceanside sequence (Wesson and Nicholson, 1988)
located at coordinates (-75,-125). The kernel associated with these events is virtually absent in the fault networks
reconstructed from 5 initial subsets (Figure 11a,b). This can be explained in terms of the atomization procedure. In the case
of 5 initial subsets, the offshore Oceanside seismicity falls in a subset containing onshore faults such as the Elsinore fault at
coordinates (0,-75). Because these faults have a more coherent spatial structure compared to the diffused Oceanside
seismicity, their proto-cluster holding capacity is higher. Hence the atomization procedure continues increasing the number
of clusters while the Oceanside seismicity has actually reached its own holding capacity. This causes nearly all of the proto-
clusters within the Oceanside region to become singular and be discarded into the background. In the case of 30 subsets, the
Oceanside seismicity is in a separate region and thus is able to retain a more reliable holding capacity estimation, yielding to
the detection of the underlying structures.
At this point, the natural question would be: which of these fault networks is a better model? The answer to this
question would depend on the application. If one is interested in the correspondence between the reconstructed faults and
focal mechanisms, or high resolution fault traces, which are expressions of local stress/strain conditions, then the ideal
choice would be the local criterion. However, if the application of interest is an earthquake forecast covering the whole
catalog domain, then one should consider the global criterion because it yields a lower BIC value, since it is formulated with
respect to the overall likelihood. We leave the statistical investigation of the fault network parameters (e.g. fault length, dip,
thickness distributions) as a subject for a separate study and instead focus on an immediate application of the obtained fault
networks.
**5. Validation through a spatial forecast test**
Several methods can be proposed for the validation of a reconstructed fault network. One way could be to project
the faults on the surface and check their correspondence with the mapped fault traces. This would be a tedious task since it
would involve a case-by-case qualitative analysis. Furthermore, many of the faults illuminated by recent seismicity might not
have been mapped or they may simply have no surface expressions. In the case of the 2014 Napa earthquake, there was also
a significant disparity between the spatial distribution of aftershocks and the observed surface trace (Brocher et al., 2015).
Another option would be to compare the agreement between the reconstructed faults and the focal mechanisms of the events
associated with them. With many of the metrics already developed (Wang et al., 2013), this would allow for a systematic
evaluation. However, the current focal mechanisms catalog for Southern California is based on the HYS-12 catalog
(Hauksson et al., 2012; Yang et al., 2012) obtained by relative double-difference techniques. As previously discussed in our
studies (Kamer et al., 2015, 2016), we have demonstrated that this catalog exhibits artificial clustering effects at different
scales. Hence, any focal mechanism based on hypocenters from this relative location catalog would be inconsistent with the
absolute locations of the KaKiOS-16 catalog.
Therefore we are left with the eventual option: validation by spatial forecasting. For this purpose, we will use the
global criterion model obtained from 30 subsets because it has the lowest BIC value of the four reconstructions presented

above. Our fault reconstruction uses all events in the KaKiOS-16 catalog, regardless of their magnitude. The last event in this catalog occurred on June 30[th] 2011. For target events, we consider all routinely located events by the Southern California Earthquake Data Center between July 1[st] 2011 and July 1[st] 2015 with magnitudes larger than M2.5. We limit our volume of interest arbitrarily to the region limited by latitudes [32.5, 36.0], longitudes [-121, -115] and depths in the range 0-20km. The likelihood scores of the target events are calculated directly from the fault network, which is essentially a weighted mixture of Gaussian PDFs and uniform backgrounds kernels. The only modification done to accommodate the forecast is aggregating all background kernels into a single cuboid covering the volume of interest. The weight of this cuboid is equal to the sum of all aggregated background kernel weights. To compare the spatial forecasting performance of our fault network we consider the simple smoothed seismicity model (TripleS) (Zechar and Jordan, 2010) that was proposed as a forecasting benchmark. This model is obtained by replacing each event with an isotropic, constant bandwidth Gaussian kernel. The bandwidth is then optimized by dividing the dataset into training and validation sets. As already pointed out by (Zechar and Jordan, 2010) the construction of the model involves several choices (e.g. choice of optimization function, choice of candidate bandwidths, etc...). To sidestep these choices we construct the TripleS model by optimizing the bandwidth parameter directly on the target set. Allowing this privilege of foresight, which would not be possible in a prospective setting, makes sure that the TripleS method is at its maximum attainable forecast skill. Figure 12 shows the forecast performances of our fault network, the TripleS model and a single uniformly dense cuboid. The performance is quantified in terms of negative log likelihood per target event for varying magnitude cut-offs of the target dataset. The reconstructed fault network performs better for all magnitude cut-off levels. We also observe a consistent relative performance increase with increasing magnitude cutoff, suggesting that the larger events tend to occur closer to the principal planes defined by the two largest eigenvalues of the fitting kernels.

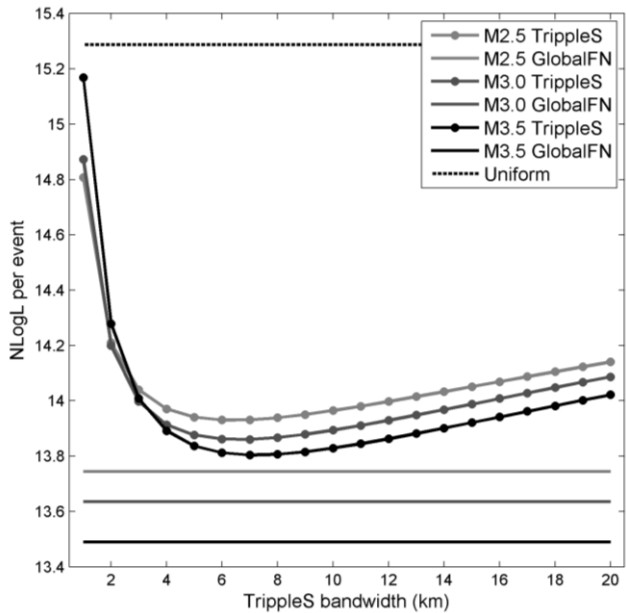

**Figure 12** Average Negative Log Likelihood for the target dataset limited to events above M2.5 (light gray), M3.0 (dark gray) and M3.5 (black). Performance of the TripleS models is evaluated as a function of the isotropic kernel bandwidth (dotted lines). The fault network performance is plotted with constant level solid lines. The performance of a single uniformly dense cuboid is plotted with a dashed line.

The superiority of our model with respect to TripleS can be understood in terms of model parameterization, i.e. model complexity. There is a general misconception regarding the meaning of "model complexity" in the earthquake forecasting community. The term is often used to express the degree of conceptual convolution employed while deriving the model. For instance, in their 2010 paper, Zechar and Jordan refer to the TripleS model as "a simple model" compared to models employing anisotropic or adaptive kernels (Kagan and Jackson, 1994, 2007). As a result, one might be inclined to believe that the model obtained by fault reconstruction presented in this study is far more complex than TripleS. However, it is important to notice that the complexity of a model is independent of the algorithmic procedures undertaken to obtain it. What matters is the number of parameters that are needed to communicate it, or in other words its minimum description length (Rissanen, 1978; Schwarz, 1978). TripleS is essentially a GMM model expressed by the 3D locations of its components and a constant kernel bandwidth. Hence it has a total of (3*479,056)+1=1,437,168 free parameters compared to the (10*385)-1=3,849 of our fault network. Thus, the difference in spatial forecasting performance can be understood in terms of the TripleS' overparametrization compared to the optimal complexity criteria employed in reconstructing the fault network. It is true that, compared to our fault reconstruction method, the TripleS model is easier to formulate and obtain. However, the fact that the isotropic TripleS kernels are co-located with hypocenters of previous earthquakes does not reduce the complexity of the model. As an everyday analogy, consider for instance an image saved as Bitmap, where each pixel is encoded with an integer representing its color: Such a representation of an image, although much simpler to encode, would require larger storage space compared to one obtained by JPEG compression. Although the JPEG compression is an

elaborated algorithm, it produces a representation that is much simpler. In the same vein, the fault reconstruction method
uses regularities in the data to obtain a simpler, more optimal representation.
Another contributing factor to the performance of the fault network can be regarded as the utilization of location
uncertainty information that facilitates condensation. This has two consequences: 1) decreasing the overall spatial entropy
and thus providing a clearer picture of the fault network, and 2) reducing the effect of repeated events occurring on each
segment, thus providing a more even prior on all segments.
**6. Conclusion**
We presented an agglomerative clustering method for seismicity-based fault network reconstruction. The method
provides the following advantages: 1) a bottom-up approach that explores all possible merger options at each step and moves
coherently towards a global optimum; 2) an optimized atomization scheme to isolate the background (i.e. uncorrelated)
points; 3) improved computation performance due to geometrical merging constraints. We were able to analyze a very large
dataset consisting of 30 years of South Californian seismicity by utilizing the non-linear location uncertainties of the events
and condensing the catalog to ~20% of its initial size. We validated the information gain of the reconstructed fault network
through a pseudo-prospective 3D spatial forecast test, targeting 4 years of seismicity.
Notwithstanding these encouraging results, there are several aspects in which the proposed methodology can be
further improved and extended. In the current formulation, the distinct background kernels are represented by the minimum
bounding box of each subset, so that they tend to overlap and bias the overall background density. This can be improved by
employing convex hulls, alpha shapes (Edelsbrunner and Mücke, 1994) or a Voronoi tessellation (Voronoi, 1908) optimized
to match the subset borders. The shape of the background kernel could also be adapted to the specific application; for
induced seismicity catalogs, it can be a minimum bounding sphere or an isotropic Gaussian since the pressure field diffuses
more or less radially from the injection point (Király-Proag et al., 2016). Different types of proto-clusters such as Student-t
kernels or copulas can be used in the atomization step or they can be introduced at various steps of the merger by allowing
for data-driven kernel choices.
The reconstructed faults can facilitate other fault related research by providing a systematic way to obtain planar
structures from observed seismicity. For instance, analysis of static stress transfer can be aided by employing the
reconstructed fault network to resolve the focal plane ambiguity (Nandan et al., 2016; Navas-Portella et al., 2020). Similarly,
the orientation of each individual kernel can be used as a local prior to improve the performance of real-time rupture
detectors (Böse et al., 2017). Studies relying on mapped fault traces to model rupture dynamics can be also extended using
reconstructed fault networks that represent observed seismicity including its uncertainty (Wollherr et al., 2019).
An important implication of the reconstructed fault network is its potential in modeling the temporal evolution of
seismicity. The Epidemic Type Aftershock Sequence (ETAS) model can be simplified significantly in the presence of
optimally defined Gaussian fault kernels. Rather than expressing the whole catalog sequence as the weighted combination of
all previous events, we can instead coarse-grain the problem at the fault segment scale, and have multiple sequences
corresponding to each fault kernel, each of them being a combination of the activity on the other fault kernels. Such a
formulation would eliminate the need for the commonly used isotropic distance in the ETAS kernels, as this single degree
kernel induces essentially the same deficiencies discussed in the case of the TripleS model. Thus, we can expect such an
ETAS model, based on a fault network, to have significantly better forecasting performances compared to its isotropic
variants.

**Code and data availability.** The Matlab implementation of the agglomerative fault reconstruction method and the synthetic
tests can be downloaded from http://www.mathworks.com/matlabcentral/fileexchange/81193 (last accessed October 2020).
The KaKiOS-16 catalog can be downloaded from http://www.ykamer.xyz/kakios/ (last accessed July 2020). The Matlab
implementation of the condensation method can be downloaded from
http://www.mathworks.com/matlabcentral/fileexchange/48702 (last accessed July 2020).

**Author contributions.** All authors conceived and designed the research. YK wrote the paper with major contributions GO
and DS. YK developed the computer codes.

**Competing interests.** The authors declare that they have no conflict of interest.
**Acknowledgments**

We would like to thank our two reviewers Leandro C. Gallo and Nadav Wetzler for their valuable comments and

suggestions, which improved this paper considerably.

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
