# Peer review of "Fault Network Reconstruction using Agglomerative Clustering: 2 Applications to South Californian Seismicity"

_Natural Hazards and Earth System Sciences, 2020_

## Referee Comment (RC1) · Nadav Wetzler (Referee) · 12 Aug 2020

The study presents a novel clustering principle to capture faults network from earthquake catalog. The authors proposes an agglomerative clustering algorithm similar to previous only here they gather clusters to faults starting with highest possible complexity level (as many kernels as possible) and gradually converge to a simpler structure.

The paper begins with a practical presentation of the problem, a failure to reconstruct a reasonable fault network based on the Landers 1992 sequence, with an unrealistic horizontal fault plain resulting the a simplify approach. The authors also address to computational limitations due to the tremendous amount of computational power (and

time) need to gather clusters with a large regional catalogs (e.g. the Southern California KaKiOS-16). In the end, authors validate the resulted fault network.

The paper is well written and figures are useful.

I have only one concern and this is the validation of the faults network. I would expect a more deterministic approach to validate the resulted faults geometry. I think that exploring a small region using FMS data + mapped fault will be much appreciated to judge, is method is more capable to detect the natural faults network?

Specific comments: • L. 33: The use of "large durations" is confusing. Please clarify. • Fig. 10: It seems that the colors of the ellipses is correlated with the size of the kernel. If so, a color bar is useful here.

---

## Referee Comment (RC2) · Leandro C. Gallo (Referee) · 13 Aug 2020

General comments

The authors propose an interesting approach by applying an unsupervised learning algorithm –agglomerative clustering- that groups together unlabeled data points to extract structural information from seismicity catalogs. Their method uses an already developed -but still relatively underused within Earth sciences- clustering technique that involves the grouping of 3D spatial distribution of seismicity according to candidate active faults. Other clustering techniques have been applied as a pattern recognition in earthquake catalogs, starting with Ouillon et al. (2008) who applied the k means

method -Ansari et al. (2009), Ouillon and Sornette (2011) and Wang et al. (2013) are some examples. The difference with other clustering techniques is that hierarchical clustering uses a bottom-up approach: each observation starts in its own cluster (having 4 points), and clusters are successively merged together. The main novelty here, therefore, is not in the clustering procedure itself, but in the bottom-up approach which is, in my opinion, a valuable step-forward towards the full understanding of natural fault network modelling. The approach is first applied to a single synthetic dataset, then to a real example featuring 3,360 points (the Landers 1992 sequence) and then, to the condensation (Kamer et al., 2015) of the KaKiOS-16 catalog (Kamer et al., 2016). The flow of the paper is globally clear, well-written and figures are suitable. That said, with the aim of making the manuscript more robust, I wish the authors had made a stronger effort to validate the application of the technique before its application to observed seismicity data. Its application to a single synthetic experiment is practical for making the whole workflow understood, however, it has no statistical significance in terms of method's sensitivity. Being that the synthetic experiment features a relatively small number of data points, I would rather advise the authors to apply the technique to a larger number of models featuring a different number of faults with diverse characteristics or orientations –without prior knowledge this would be computationally inexpensive. Assessing discrepancies between the true and the inferred plane segments in a number large enough would then allow statistically meaningful results that, in my opinion, would make the whole manuscript more robust. I will not insist to modify the current version, but I would urge to at least think about this before you send this off to the printing press. Having said that, I can recommend publication after some modification, as I believe that the aims and approach are valid and also has relevance to a number of applications within the geosciences (esp. seismic hazards and structural geology/geomatics, but also to the extraction of planar facets in digital outcrop models).

Specific comments

L. 52: The contribution of source code to this section as supplementary materials –or

open-access code repositories like GitHub or Zenodo- would boost scientific progress and reproducibility. L. 53: I don't see this subsection appropriate for the "methods" section. L.86-88: The criterion applied for merging two clusters involves the minimum squared Euclidean distances, was this criterion chosen for any particular reason? Is there any other metric to use instead for clustering? I'm thinking about the Eigen-based parameters of the covariance matrix. It would be valuable some extra explanation. L. 110, Figure 2: for those who are unfamiliar with the method, the hierarchical, binary cluster tree is most easily understood when viewed graphically. It would be helpful for the understanding of those who are not familiar to add the associated dendrogram to this figure.

---

## Author Comment (AC1) · 15 Oct 2020

Thank you for your time in reviewing our paper. Below is our response to your comments and description of the modifications we made to address them.

*I would expect a more deterministic approach to validate the resulted faults geometry. I think that exploring a small region using FMS data + mapped fault will be much appreciated to judge, is method is more capable to detect the natural faults network?*

We agree with the sentiment that focal mechanism solutions (FMS) can reveal important information regarding the geometry of the faults under investigation. Deterministic measures that investigate the compatibility between reconstructed faults and the focal mechanisms of the events have been introduced and extensively studied by our group in research by Wang et al 2013. The FMS are greatly dependent on the location and the velocity model used for the inversions. Thus it would be inconsistent to use the solutions obtained using cross-correlation based relative locations with our absolute location catalog obtained using a different velocity model. That is why we focused on validations based on information criteria and cross correlation.

In an effort to address your comment we updated Figure 5 (now Figure 6) by adding the fault trace of the Landers fault as obtained from the Community Fault Model of southern California. We also added the following text to inform the reader about FMS based validation approaches.

*"It is also possible to employ metrics based on consistency of focal mechanism solutions to evaluate the reconstructed faults. For a detailed application of such metrics the reader is referred to the detailed work by Wang et al.(2013). In this study, since we do not have focal mechanism solutions for our target catalog, we focus on information criteria metrics and out of sample forecast tests."*

**Specific comments: L. 33: The use of "large durations" is confusing. Please clarify. Fig. 10: It seems that the colors of the ellipses is correlated with the size of the kernel. If so, a color bar is useful here.**

We have clarified the term to express catalogs covering long time spans. In Figure 10 (now Figure 11) have added a color bar and also supplemented the figure caption to indicate that the color axis unit.

*""Clusters are colored according their density (data point per km3) where the volume is estimated as the product of standard deviations along the principal component axes."*

**References:**

Wang, Y., G. Ouillon, J. Woessner, D. Sornette, and S. Husen (2013), Automatic re-construction of fault networks from seismicity catalogs including location uncertainty, *J. Geophys. Res. Solid Earth, 118(11)*, 5956–5975, doi:10.1002/2013JB010164.
* * *

---

## Author Comment (AC2) · 15 Oct 2020

Thank you for taking the time to review our paper and providing detailed suggestions. Below is our response to your comments and description of the modifications we made to address them.

*Its application to a single synthetic experiment is practical for making the whole workflow understood, however, it has no statistical significance in terms of method's sensitivity. Being that the synthetic experiment features a relatively small number of data points, I would rather advise the authors to apply the technique to a larger number of models featuring a different number of faults with*

[Figure]

*diverse characteristics or orientations –without prior knowledge this would be computationally inexpensive. Assessing discrepancies between the true and the inferred plane segments in a number large enough would then allow statistically meaningful results that, in my opinion, would make the whole manuscript more robust.*

We agree with you that the synthetics provided previously did not allow for conclusions about the sensitivity and robustness of the method. We have therefore supplemented the synthetics section with a more elaborate study where we gradually increase the sampling of a ground truth fault network under different background noise levels and investigate the method's clustering performance using the Rand index. This is now covered in section 3.1 and the results are provided in Figure 5.

*L. 52: The contribution of source code to this section as supplementary materials –or open-access code repositories like GitHub or Zenodo- would boost scientific progress and reproducibility.*

Based on your suggestions we have made publicly available the codes for the agglomerative clustering and the codes for the generation and evaluation of the synthetic sensitivity analysis here: https://www.mathworks.com/matlabcentral/fileexchange/81193. The link is included in the "Code availability" section.

*L. 53: I don't see this subsection appropriate for the "methods" section.*

We have moved this part to a separate section after the introduction.

*L.86-88: The criterion applied for merging two clusters involves the minimum squared Euclidean distances, was this criterion chosen for any particular reason? Is there any other metric to use instead for clustering? I'm thinking about the Eigen-based parameters of the covariance matrix. It would be valuable some extra explanation.*

Our selection of the Ward's criterion was motivated by its characteristic of producing

regular sized clusters. This is important for the atomization procedure because we want all clusters to have the same potential to merge and grow into bigger structures. Initially we also investigated using the Mahalanobis distance with single linkage, and using the Gaussian associated with the location uncertainty of each event without atomization. These methods were not successful in reconstructing the synthetic networks in the presence of background noise; hence we focused our attention on atomization using the Ward creation. We have added the following sentence to the method section.

*"While there are many different linkage methods and distance metrics, here we have chosen to use the Ward's criterion because it produces clusters with regular sizes. This is important for the atomization procedure as we want clusters to have similar potentials to merge and grow into bigger structures."*

**L.110, Figure 2: for those who are unfamiliar with the method, the hierarchical, binary cluster tree is most easily understood when viewed graphically. It would be helpful for the understanding of those who are not familiar to add the associated dendrogram to this figure.**

We have added the dendrograms for both datasets to Figure 2.